# Ultimate Strength Study of Structural Bionic CFRP-Sinker Bolt Assemblies Subjected to Preload under Three-Point Bending

**DOI:** 10.3390/biomimetics8020215

**Published:** 2023-05-23

**Authors:** Zhengqi Qin, Ying He, Shengwu Wang, Cunying Meng

**Affiliations:** 1School of Mechanical Engineering College, Dalian Jiaotong University, Dalian 116028, China; 2School of Transportation and Electrical Engineering, Dalian Institute of Science and Technology, Dalian 116052, China; 3College of Aerospace Engineering, Shenyang Aerospace University, Shenyang 110136, China

**Keywords:** bolt preloading force, countersunk head bolted, bending load

## Abstract

Countersunk head bolted joints are one of the main approaches to joining carbon fiber-reinforced plastics, or CFRP. In this paper, the failure mode and damage evolution of CFRP countersunk bolt components under bending load are studied by imitating water bears, which are born as adult animals and have strong adaptability to life. Based on the Hashin failure criterion, we establish a 3D finite element failure prediction model of a CFRP-countersunk bolted assembly, benchmarked with the experiment. The analysis shows that the simulation results under specified parameters have a good correlation with the experimental results, and can better reflect the three-point bending failure and fracture of the CFRP-countersunk bolted assembly. Based on the specified parameter of the carbon lamina material change, we used the countersunk bolt preload to investigate the stress distribution near the counterbore zone, and to investigate the effect of bolt load on the three-point bending limit load. The results obtained using FEA calculations indicate that the stress distribution around the countersunk hole is related to the laminate direction. The bolt preloading force increasing reduces the load value at the initial damage, and the appropriate preload force will increase the ultimate load of the joint.

## 1. Introduction

Water bears are the most biologically viable creatures known on Earth, surviving in outer space without protection, in the Himalayan mountains (above 6000 m, once found at 5546 m), in hot springs, in Antarctica and in the deep sea (below 4000 m). Their primary feature is that they is born from eggs and in adult form, and the number of cells in their bodies does not change throughout their life (without childhood). The water bear has a strong ability to adapt to the environment, and this depends on the mature and capable cells that it has from birth. Composite bolted joints in aerospace are the weak points of the overall structure, and their ultimate stress will affect the applicability of the overall structure. Changes in bolt preloads affect the ultimate stress of the composite structure’s bolted connections; if this part is similar to the water bear (i.e., at the beginning of the design and manufacturing, its ultimate stress can reach the maximum preload), then the composite bolt connection will also have super strength similar to that of the water bear, as well as better usability. At the same time, this may reduce the probability of component failure, the structure may be safer, and at the same cost, bear larger loads.

The special mechanical properties exhibited by organisms in nature to adapt to their living environment have inspired the development of new materials and structures. Among them, carbon fiber-reinforced plastics (CFRP) are widely used in the bionic, aerospace, automotive, and other fields due to their excellent mechanical properties [1,2]. In the engineering field, CFRP components are often used to support load transitions, usually using two types of CFRP connections, namely bonded and mechanical connections [3]. Bolted connections, as a type of mechanical connection, have gained special attention in aircraft applications due to their strong ability to withstand external loads, and their low dispersion of connection strength.

Several authors have studied CFRP-countersunk head bolted structures. Zhai et al. [4] investigated the effect of joint interface conditions on the bearing response of single-lap, countersunk composite–aluminum bolted joints. Their experimental results showed that the joint interface conditions had a significant effect on the bearing response of the joints, with the bonded joints exhibiting a better bearing response compared to the unbonded joints. They also investigated the influence of bolt-hole clearance and bolt torque on the strength of single-lap, countersunk composite joints through an experimental study [5]. The results indicate that an optimal combination of clearance and torque can maximize the strength of the joints. However, excessive clearance or torque can decrease the strength. McCarthy and Gray [6] presented an analytical model for predicting load distribution in highly torqued multi-bolt composite joints. The model was validated by comparing its predictions to the experimental results. The study concluded that the model is an accurate and efficient tool for predicting load distribution in composite joints under high torque levels. Kolks and Tserpes [7] proposed an efficient progressive damage model for hybrid composite/titanium bolted joints. The study concluded that the model is capable of accurately predicting the failure of hybrid composite/titanium bolted joints under various loading conditions. Longquan Liu et al. [8] investigated the combined and interactive effects of interference fit and preloads on composite joints. The study concluded that interference fit and preload have significant effects on the mechanical behavior of composite joints, and that the optimal combination of these parameters can improve the joint’s load-carrying capacity. Mandal and Chakrabarti [9] conducted numerical simulations to evaluate the failure of multi-bolt-composite joints with varying sizes and preloads of bolts. The study found that increasing the preload on the bolts improved the strength of the joints, and that the size of the bolts had a significant effect on the joint’s failure mode. Chishti et al. [10] conducted a numerical analysis to investigate the damage progression and strength of countersunk composite joints. The results showed that the damage in the joints initiated near the hole edges and gradually spread towards the outer regions of the joint, leading to eventual failure. Qingbo Liu et al. [11] explored the impact of preload on the tensile properties of composite single-lap joints. The experimental findings indicate that increasing the preload enhances joint strength, and the failure mode transitions from adhesive failure to cohesive failure. Nonetheless, excessive preload has the potential to damage the composite material. SENF et al. [12] conducted an experimental study on the failure of mechanically connected joints of composite laminates with gaps under preload. The results showed that the preload had a significant effect on the failure mode and load-carrying capacity of the joints. An appropriate preload can improve the mechanical properties of the joint, but an excessive preload may lead to damage to the joint. Calin et al. [13] presented a numerical analysis of the effects of geometrical and mechanical parameters on the stiffness and strength of composite bolted joints. The results show that the most influential parameters are the bolt diameter, the laminate thickness, and the material properties. Bodjona K et al. [14] presented a model that can predict the load-sharing behavior of single-lap bonded/bolted composite joints. The model is validated through experimental tests, and the results show good agreement between the predicted and measured load-sharing behavior. Liu, F. [15] proposed a two-dimensional finite element model to predict the load distribution in bolted joints. The model uses shell elements and fastener techniques to simulate laminates and bolts. By improving the simulation method of bolt stiffness, adding bolt holes and selecting appropriate influence areas, it can better predict secondary bending effects. The author compared the improved model with other models and three-dimensional models to validate that the model improves efficiency while improving prediction accuracy, providing an optional method for failure analysis and the design of bolted joints. Valerio G. Belardi et al. [16] applied a new composite bolted joint element (CBJE) to FE analysis of single-lap, single-bolt joint. It represents the elasticity of the joint and surrounding plates using radially beam elements with stiffness from the authors’ theory model solution. For single-lap joints, a 3D full model comparison with a shell model using CBJE shows CBJE improves both efficiency and accuracy. Additionally, it offers an optional accurate and efficient method for composite bolted joint analysis.

However, most studies on the tensile performance of CFRP-sunken bolt assemblies have simplified the loads and boundary conditions and focused only on the mechanical properties of the joint under tensile resistance. Nevertheless, since structures in actual engineering are often dominated by complex operating conditions, the over-simplified load forms may not effectively guide the research results of engineering design.

Therefore, it is necessary to further investigate the various stress conditions in the joints, especially the load distribution in the bolt holes and the effect of preload on the bending strength of single countersunk bolt-CFRP assemblies. The purpose of this study is to validate the experiments using a simulation method based on Hashin’s damage criterion, and to investigate the aforementioned issues.

The simulation method based on Hashin’s damage criterion was used to predict the damage and failure behavior of the composite material under various loading conditions. The simulation model consists of a two-dimensional model with a circular hole and a single countersunk bolt. The effect of preload on the bending strength of the single countersunk bolt-CFRP assembly was investigated by analyzing the stress distribution in the bolt holes. The effect of bolt preload on component deformation was also investigated to assess its effect on the bending strength.

In summary, this study aims to expand knowledge of the mechanical behavior of single countersunk bolt-CFRP assemblies under different loading conditions. The simulation approach based on the Hashin damage criterion is expected to provide a comprehensive understanding of joint mechanics. The results of this study have the potential to guide the engineering design and development of new CFRP structures with enhanced mechanical properties.

## 2. Experimental Approach

### 2.1. Experimental Tests

#### 2.1.1. Loading Test System

The current study employs a state-of-the-art 50-ton multi-axis fatigue electro-hydraulic servo test system that has been developed in-house, as illustrated in Figure 1. This sophisticated testing apparatus can conduct several types of mechanical tests, including uniaxial dynamic stretching, biaxial dynamic stretching, uniaxial tensile static/cyclic test, and three-point bending static/fatigue tests. The equipment is a universal electromagnetic resonance fatigue testing machine, which can perform tension-compression fatigue tests on metallic materials below 250 Hz. The introduction of microcomputer control technology has expanded the functions of traditional high-frequency fatigue testing machines. Computer technology automatically calibrates load amplifiers, stabilizes load control, and records test data during the fatigue process. It also provides printing reports under different work modes and displays various operation tips through the CRT. The electrical control system uses advanced pulse width modulation and switch-type transistor power amplifiers, making it easy to operate with high efficiency. The average load control system is directly driven by a rotating motor which loads and unloads quickly with high precision. The mainframe system adopts a five-degrees-of-freedom mechanical model, with more uniform distribution of force than other similar products. The device is adjusted well at the factory and is easy to operate. The development of this test system has significantly contributed to our ability to evaluate the mechanical properties of materials accurately.

The three-point bending test was selected as the testing method for this study. This test involves applying cyclic loading to a sample until failure, simulating the bending stresses that a material may face in its practical applications. In this experiment, the sample is subjected to a displacement loading of 10 Hz, with minimum and maximum displacement values of 0 mm and 1 mm, respectively. Additionally, the displacement amplitude was maintained at 1 mm throughout the test. These testing parameters were chosen based on their ability to simulate realistic working conditions and provide precise measurements of the material’s performance.

#### 2.1.2. Loading Head and Support Design

In this experiment, the loading head is a fixed loading head. It is guaranteed that the loading head can ensure the reliability of its long-term periodic loading when clamped to the test machine. The radius of the loading head is r = 10 mm, the width is 50 mm, the height is 20 mm, the machine clamping position is 50 mm high, and the thickness is 5 mm, as shown in Figure 2. The indenter material is made of 45 steel, the hardness requirement is 60~62 HRC, and it has a delicate bottom surface, no nicks, no burrs and trimmed sharp edges; the surface roughness is Ra 6.3, and the bottom roughness adopts Ra 3.2 to prevent damage to the indenter surface of the specimen.

The support is made of rolling cylindrical smooth shaft standard parts, themselves made of chrome-plated 45 steel; the support’s hardness meets the test requirements. The surface roughness is Ra 0.3~0.6. The bracket is shown in Figure 3.

A fixture is required to complete the positioning and clamping function of the test piece. The design of the fixture is designed according to the shape of the rolling support. The whole base can be restricted from the front and back of the test piece, and clamped as shown in Figure 4.

Within the base design, the rolling support is subjected to the downward force of the carbon fiber board, and the contact surface of the base and the rolling support is designed as φ 10 round surface. The advantage of this design is that the base can fully absorb the force from the rolling support, so that the force of the support is more uniform, which is equivalent to improving the use strength of the support. In addition, the round surface enables the rolling support to roll. When the specimen is deformed, the support roll can prevent the specimen experiencing damage. A sliding table is placed at the bottom of the base to facilitate the clamping sliding, which makes it easy to change the span of the carbon fiber board bending test. A φ 10 through-hole on the outside side of the base is milled out with a T-bolt lock, as shown in Figure 5. The base material of Q235 is selected, milled, and processed. In order to avoid rust, the surface is treated with blue. In roughness, the whole fixture is Ra 6.3, and it has general tolerance.

The card limits the axial position of the rolling support. The upper and lower lock blocks mainly clamp the upper limit shaft and the upper limit shaft, and the height of the upper limit shaft can be controlled by the mid-hole, and the top hole is connected with two M5 screw. Under the front and rear limit plates, the test piece is limited through two nosops, and the mid-bar can be used with different widths. The material is Q235; it undergoes wire-cutting processing and surface blue treatment, has a roughness Ra 6.3, and has a general tolerance.

The slide rail is connected with T-bolts to adjust the span, and the bottom side is clamped with a support column on the test system. The slide material is 45# steel with excellent performance in all aspects, and the unity of the slide clamp is guaranteed through washing processing to ensure the strength and stiffness of the fixture. In order to facilitate the screw rotation, 0.5 mm chamfer is poured on the specific screw hole, the surface roughness of the whole slide rail is Ra 3.2, and the tolerance is general tolerance. The slide rail is shown in Figure 6.

The support column material is 45# steel, with excellent performance. The bottom of the support column is cut into two planes to remove the wrench; an M10 thick tooth thread is connected to the guide rail by adding an M10 elastic pad to prevent looseness, and the rest of the chamfer is 0.5 mm 45°, as shown in Figure 7.

The bent loading head is installed on the plane loading fixture above the testing machine and clamped through the ratchet wrench, as shown in Figure 8.

#### 2.1.3. Test Specimen Preparation and Load Application

The specimen material presented in this article is T300BZ-3234(Hengshen Co., Ltd., Danyang, China.). Table 1 shows the sample materials. The countersunk head bolts are M6 bolts (according to the Chinese navigation mark “HB/1-129-2002 90° countersunk head bolts”) with material Ti6Al4V. A countersunk hole is drilled in the middle of the laminate plate with CVD diamond-coated tools to ensure the machining quality. Additionally, an M6 countersunk head bolt was placed through the hole with a steel washer and nut, these were then placed under the bottom plate, as illustrated in Figure 9.

### 2.2. Loading and Test Results

Upon completion of the test, it was observed that the upper plate of both sample 1 and sample 2 did not fracture completely, while the lower plate fractured completely, as shown in Figure 10a. This fracture pattern indicates that the material has high fracture toughness, which is a desirable property for materials subjected to high stresses or loads over a long period of time. Figure 10b shows the force–displacement curve of the loading end during the three-point bending test. The abscissa represents the displacement, the ordinate represents the force, the black curve is the test piece 1, and the red curve is the test piece 2. The curves of the two groups of specimens have a similar bending stiffness and fracture load; it can also be seen that the specimens experience sudden fractures.

## 3. Failure Analysis of CFRP-Countersunk Bolt Assembly

### 3.1. Hashin Damage Criterion

In order to comprehensively describe the various damage scenarios during the bending damage to composite materials, the Hashin damage criterion was chosen for progressive damage analysis [15]. The Hashin three-dimensional damage criterion is capable of predicting four types of damage, including fiber tensile damage, fiber compression damage, matrix tensile damage, and matrix compression damage. This criterion provides a valuable tool for predicting and analyzing the failure modes of composite materials, which is essential for improving the design and performance of these materials in various applications.

This numerical simulation uses ABAQUS, with preloading of Altair Hypermesh. The Hashin criterion is integrated in ABAQUS. Additionally, is used to describe the interaction between damage and composite materials.

### 3.2. Geometry and Mesh Model

The simulation used in this study accurately reflects the dimensions of the physical specimen being tested. The equivalent bolt diameter in the finite element (FE) model is 5.3 mm, with the bolt and nut being modeled as common nodes. The gasket is represented by a washer with an inner diameter of 5.3 mm, an outer diameter of 10.06 mm, and a thickness of 1.5 mm, as illustrated Figure 11. The pivot point and the loaded indenter are treated as discrete rigid bodies in the model, and their mechanical properties are not taken into account. This approach enables the simulation to capture the mechanical behavior of the specimen and the bolted connection under specific loading conditions such as tension or compression. The composite laminate uses SC8R continuous shell cells with reduced integration, and the global cell size is 0.3 × 0.3 × 0.1 mm, corresponding to the laminate thickness direction. There are a total of 30 layers of cells; the cell layup 0° points along the *x*-axis, the layup 90° points along the *y*-axis, the countersunk bolts and spacers use C3D8R cells with reduced integration, and the global cell size is 0.33 × 0.33 × 0.2 mm, of which the size along the laminate thickness direction is 0.2 mm. The global cell size is 0.33 × 0.33 × 0.2 mm, including 0.2 mm along the thickness direction of the plywood, and discrete rigid body modeling is used for the pivot points and ends, with a mesh size of 0.2 × 0.2 mm, as illustrated in Figure 12.

### 3.3. Material Properties

Table 1 and Table 2 provide a detailed list of the material properties of the countersunk head bolt, gasket, and composite laminas. These properties are essential in predicting the mechanical behavior of the components during bending load and subsequent material damage. In order to comprehensively describe the various damage situations that may occur in the damage process, the Hashin damage criterion was selected for the analysis of composite materials [17]. Additionally, Table 3 and Table 4 provides the friction coefficients of specific contact pairs [18], which are crucial in accurately modeling the contact behavior between the different components of the test.

### 3.4. Load and Boundary Conditions

To maintain consistency with the test procedure, the numerical model was divided into two steps. In the first step, a preload force was applied to the model to study the stress distribution around the nail hole when only the preload force was applied. In the second step, a forced displacement boundary was applied to the rigid indenter of the model to simulate the three-point bending of the specimen. This loading process can accurately reflect the test loading process.

(1)Appling bolt preloading force

Identical bolt preload torque is related to preload force, according the literature [19]:(1)$$T=KF^{\prime }d$$
where *T* represents the preload torque. *d* denotes the thread diameter. *F′* is the preload force. *K* is the tightening torque coefficient.

For the M6 bolt, *d* equals 6.02 mm. *K* varies from 0.1 to 0.3, and *K* = 0.2. Applying the preload using the cooling method, that is
(2)$$F^{\prime }=E\times A\times \alpha_{3}\times \Delta t$$
where *E* represents the Young’s modulus, and *A* is the cross-section area. $\alpha_{3}$ is the linear expansion coefficient by the axial direction of the bolt. ∆*t* is the cooling range.

Based on the Equation of (1) and (2), ∆*t* = 1130 °C.

As illustrated in Figure 11, the *x*-axis points in the direction of length, and the *z*-axis points in the direction of loading. Due to the symmetry of the mode, only half of the entire model is considered in the longitudinal direction, meaning that the symmetric surface is the XZ-plane crossing the center line of the bolt, and the symmetric boundary conditions are applied to the symmetric plane. Specifically, the y-direction displacement of the symmetric surface is fixed, while the x and z directions are released. Moreover, the rotation around the *x*-axis and *z*-axis of the symmetric surface is fixed, while the rotation around the *y*-axis is released. This modeling approach significantly reduces the computational cost, while still allowing accurate predictions of the behavior and performance of the structure under specific loading conditions.

(2)Three-point bending

In this study, a displacement load of 4 mm is applied downward to the pivot while considering the pivot and indenter as rigid body constraints, which simplifies the analysis process. The study aims to determine how the structure will react under a given load, and how the stress and strain will be distributed throughout the structure.

### 3.5. Meshing

The model meshing in this paper can be found in the literature [17].

### 3.6. Analysis Results

Figure 13, Figure 14, Figure 15 and Figure 16 depict the damage contour maps of the matrix and fibers, which show a similar failure mode to that of Figure 10a. These results indicate that the CFRP-countersunk head bolt structure exhibited matrix compression damage on the top plate and fiber and matrix tensile damage on the bottom plate under a three-point bending load. In addition, minor fiber compression damage was observed on the top plate. However, the damage results of the simulation analysis were not entirely symmetrical on both sides of the specimen’s axis of symmetry, as observed from the top view of the specimen. The damage values on the left side of the specimen were greater for both fiber tensile/compression damage and matrix tensile/compression damage. These results have practical implications for the design and optimization of composite structures, as well as for enhancing our understanding of the failure mechanisms and performance of composite bolted joints. Figure 17 illustrates the force–displacement curve for the loading process using the FEA model (green curve), which was extracted and compared to the test results. The force–displacement curve of the simulation analysis is similar to the experimental curve in terms of bending stiffness, with an error in fracture load of 5.2%. The correlation between the simulation and experiment results shown in Figure 17 indicates that the material parameters used in the simulation are reasonable.

## 4. Effect of Preload on Bending Ultimate Load

### 4.1. Counterbore Stress Distribution

The present study applied preloads of 2.0 kN, 4.0 kN, 6.0 kN, and 8.0 kN to the FEA model using the parameters specified earlier. Based on Equations (1) and (2), the temperature changes were computed as −512.84 °C, −1025.68 °C, −1537.74 °C, and −2051.36 °C, respectively. The stress distribution obtained from the analysis is illustrated in Figure 17 and Figure 18.

The stress distribution around the countersink displayed in Figure 18 is not uniform, and is related to the ply orientation. The periphery experiences more stress in the direction parallel to the fiber. To demonstrate this stress behavior further, the circumference of the bottommost circumferential node of the countersink slope was taken as the stress path, as illustrated in Figure 19. The path’s total length was set to 1, and the maximum principal stress value on the path was extracted. The relative length of the maximum principal stress (MPa) along the stress path is shown in the figure. The results indicate that the maximum principal stress value first increases sharply and then slowly. The stress level at the slope of the countersink is higher than that at the bottom of the countersink, suggesting that the slope of the countersink is a critical location with high stress concentration. This information can be utilized for optimizing the design and enhancing the reliability of the bolted joints. Further research can investigate the effects of different factors on the stress distribution around the countersink.

The maximum principal stress increases significantly with the increase in the preload in the direction parallel to the fiber, and the stress value is almost linear with the increase in the preload, while the stress change in other directions is not significant. This phenomenon occurs because the Young’s modulus of the laminate parallel to the fiber direction is higher than that of other directions. Therefore, with the increase in preload force, the difference in the amount of deformation in the parallel to the fiber direction and other directions becomes larger, leading to a larger stress difference. To prevent material damage and the impact on the performance of the connection structure, it is essential to avoid applying excessive preload force in actual engineering. Excessive local stress can occur due to the application of excessive preload force, which can have an adverse effect on the connection structure’s reliability.

### 4.2. Effect of Preload on Three-Point Bending

Four sets of analysis models with preloads of 2 Nm, 4 Nm, 6 Nm, and 8 Nm were established, respectively. A bending load of 4 mm was applied to the specimens and the displacement–load curve of the specimen was obtained, as shown in Figure 20. The results show that within the given range of preload forces, the preload force improves the bending stiffness of the specimen to a certain extent, but the ultimate load does not increase with the increase in preload force. According to the results, there exists an optimal preload force that maximizes the ultimate load, beyond which the ultimate load decreases. This conclusion is similar to that given in reference [9]. In these four sets of analyses, the optimal preload force is 4 Nm, and the corresponding ultimate load is 2.41 kN.

## 5. Conclusions and Outlook

Taking the super ability of biological tardigrades to adapt to the environment as a bionic model, the influence of the CFRP-countersunk bolt structure on the ultimate stress of the component under preload and bending load was studied, and corresponding conclusions were drawn. The results are as follows:

(1) According to research on CFRP-countersunk bolted structures, the stress distribution around the countersunk hole is affected by the direction of material orientation, and the maximum principal stress increases with preload. This suggests that these factors should be carefully controlled to prevent stress concentration and potential failure in the component.

(2) The application of preload force can improve the bending stiffness of the connector. However, excessive bolt prestress can lead to an increase in material stress around the hole, which in turn decreases the ultimate bending load. Therefore, it is necessary to investigate the effect of bolt preload on the mechanical behavior of CFRP-countersunk bolted structures to enhance understanding of joint mechanics and provide guidance for the engineering design of new CFRP structures.

(3) In CFRP-sunken head bolt connection structures, the optimum value of preload is essential to achieve the maximum bending load subjected to the preload. Therefore, it is important to investigate the relationship between the magnitude of the preload force and the connection structure itself. An appropriate value of preload can significantly increase the bending stiffness of the connection, while an excessive bolt preload can lead to an increase in material stresses around the hole and a decrease in the ultimate bending load. Further studies are needed to understand the influence of different factors, such as material properties, geometric parameters and loading conditions, on the optimum preload values for CFRP-sunken head bolted connection structures.

Biomimicry is broad and profound, and the application of bionics can solve thorny problems in engineering and provide intelligent theories from nature for the development of human beings and the creation and renewal of engineering. In this paper, the influence of the preload force of CFRP countersunk bolt joints on ultimate stress is studied by imitating the ability of water bears to adapt to the world. A variety of factors determine that water bears have a strong ability to adapt to the living environment. Therefore, future research beyond this paper may also study different load conditions, such as low temperature environment, joint load, etc. Combined with different influencing factors, components that are more suitable for the environment can be designed to contribute to the development of the aerospace industry.

## Figures and Tables

**Figure 1 biomimetics-08-00215-f001:**
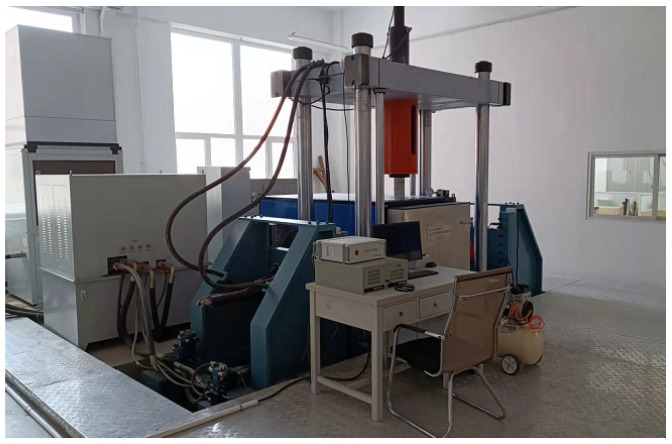
Test system.

**Figure 2 biomimetics-08-00215-f002:**
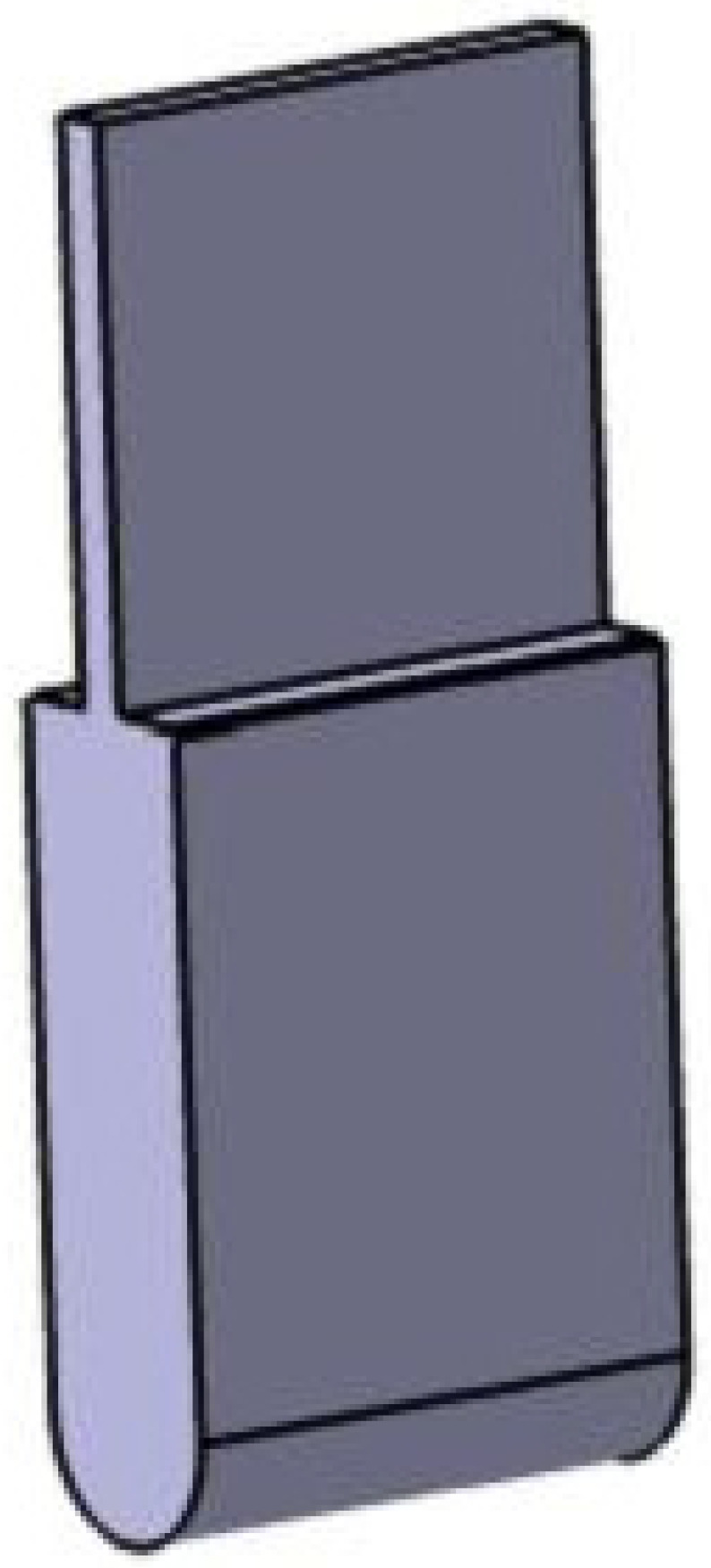
3D diagram of the loading head.

**Figure 3 biomimetics-08-00215-f003:**
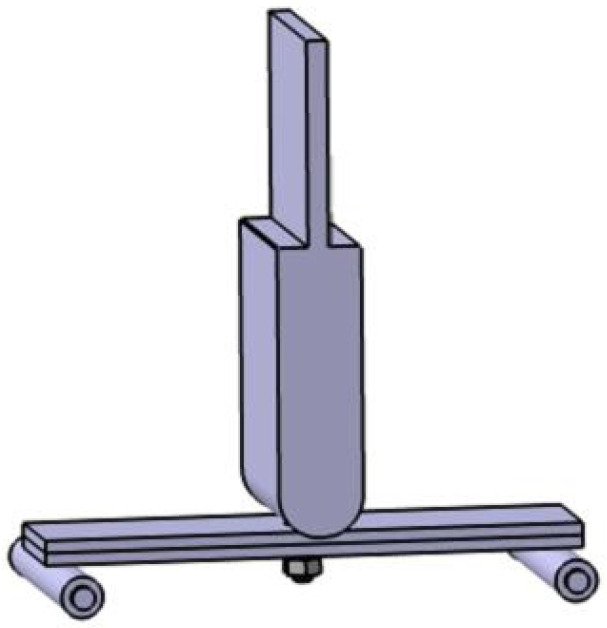
Three-point loading mode, with fixed loading head and rolling support.

**Figure 4 biomimetics-08-00215-f004:**
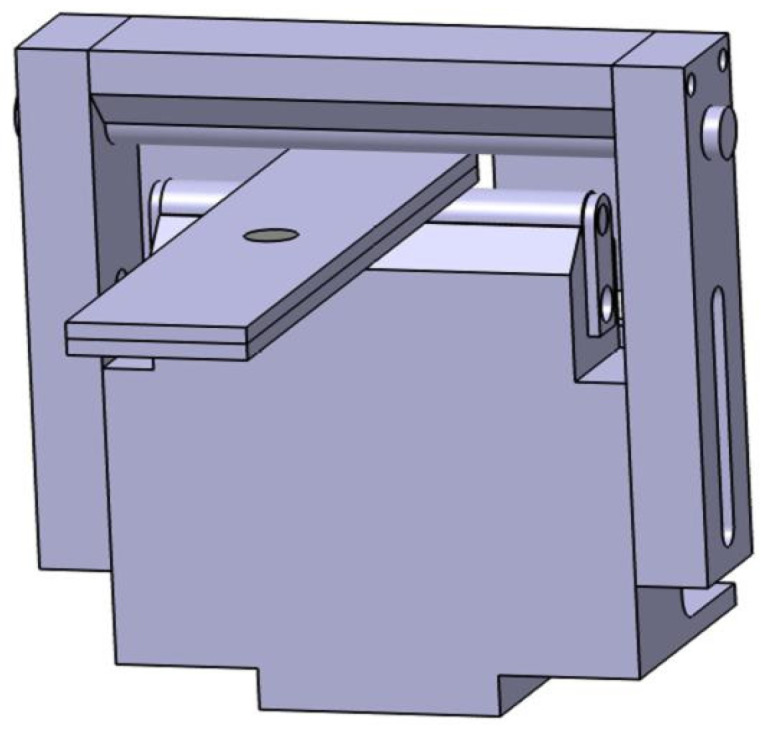
Left base fixture.

**Figure 5 biomimetics-08-00215-f005:**
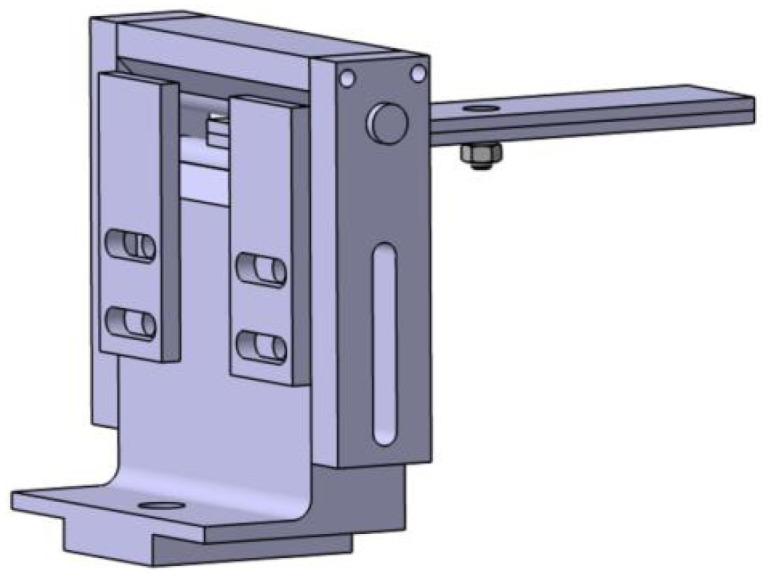
Left base fixture.

**Figure 6 biomimetics-08-00215-f006:**
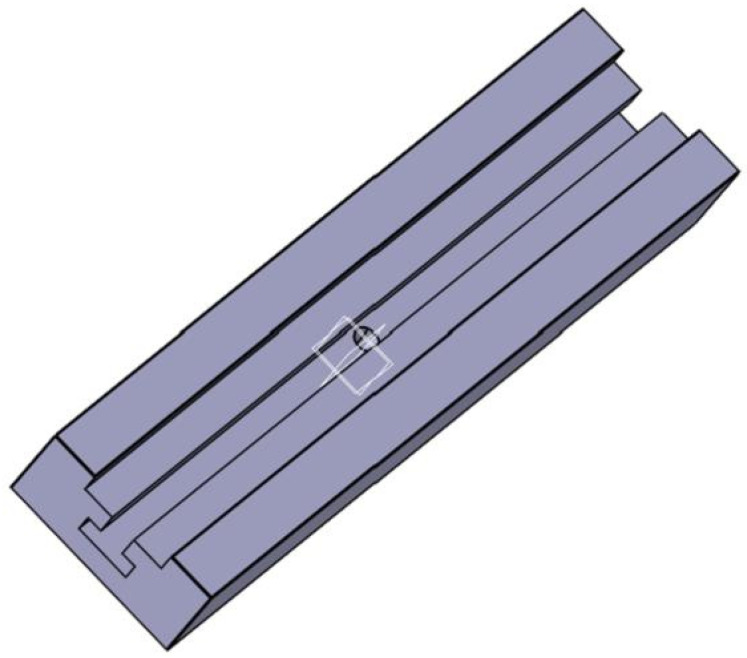
Slide rail.

**Figure 7 biomimetics-08-00215-f007:**
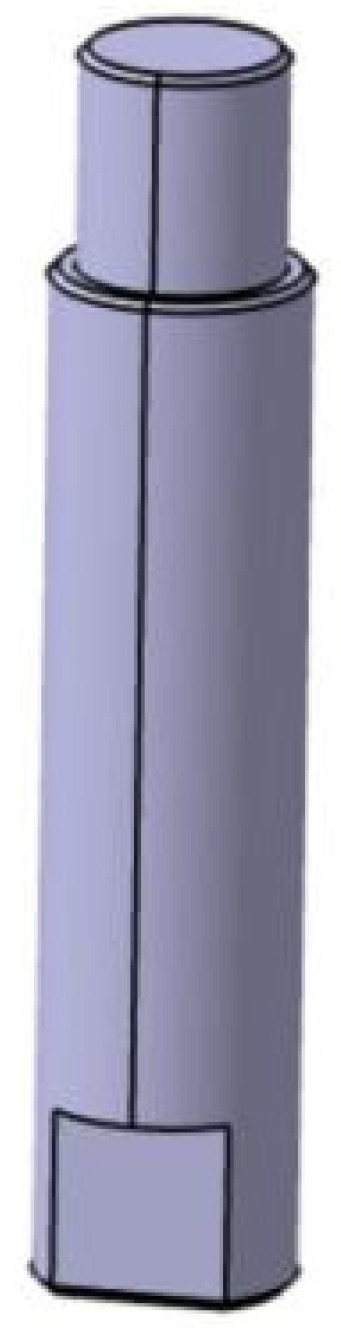
The support column.

**Figure 8 biomimetics-08-00215-f008:**
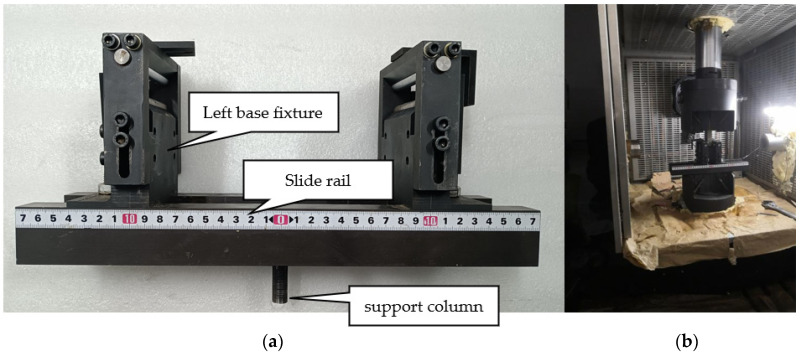
Fixture and installation. (**a**) Fixture structure (**b**) Installation formal.

**Figure 9 biomimetics-08-00215-f009:**
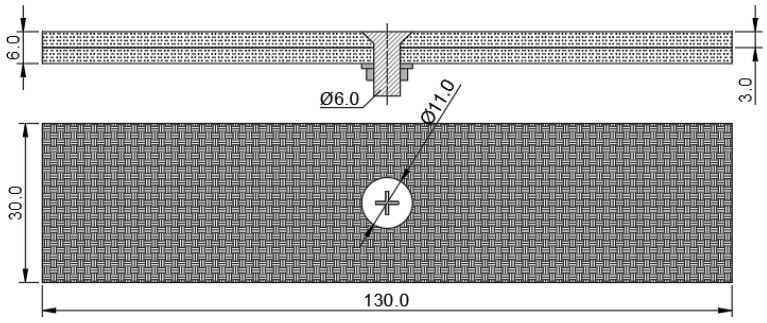
Countersunk bolt laminate specimen.

**Figure 10 biomimetics-08-00215-f010:**
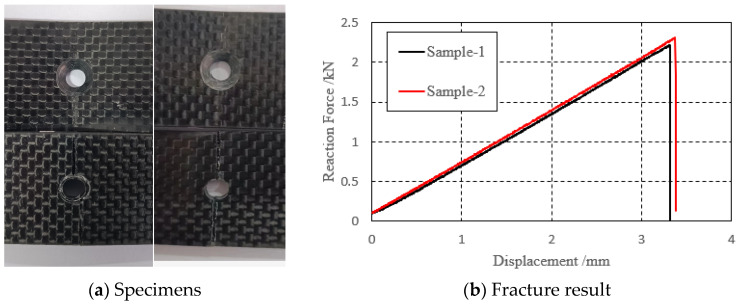
Specimens fracture result.

**Figure 11 biomimetics-08-00215-f011:**
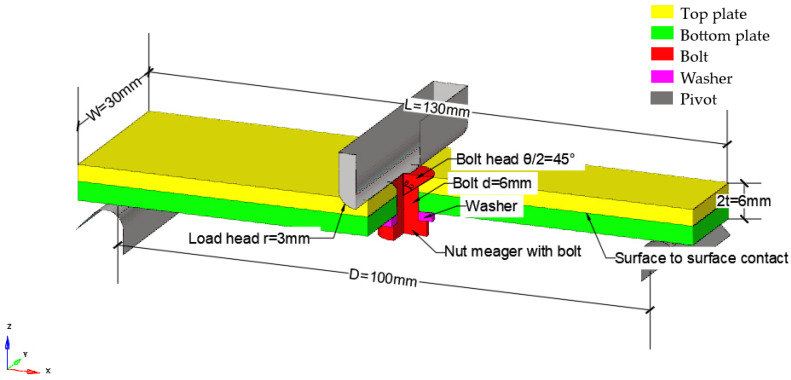
Schematic diagram of the geometry.

**Figure 12 biomimetics-08-00215-f012:**
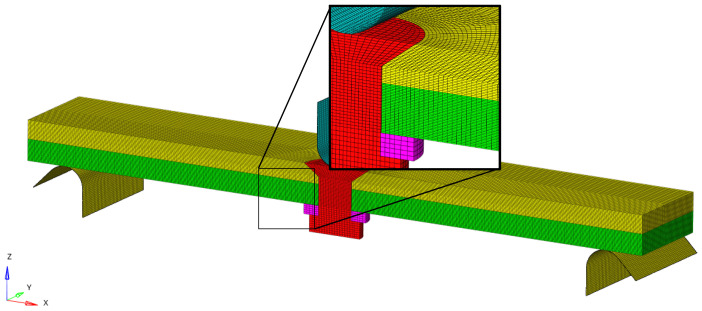
FE model mesh distribution.

**Figure 13 biomimetics-08-00215-f013:**
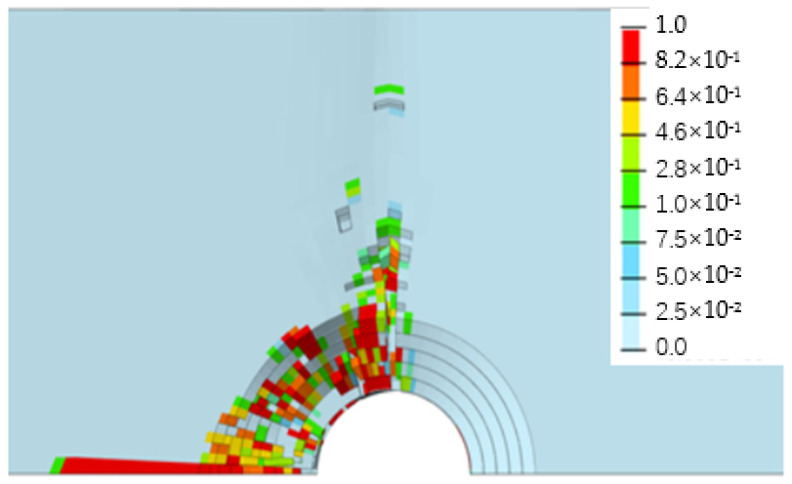
Fiber compression damage on the upper surface of the top plate.

**Figure 14 biomimetics-08-00215-f014:**
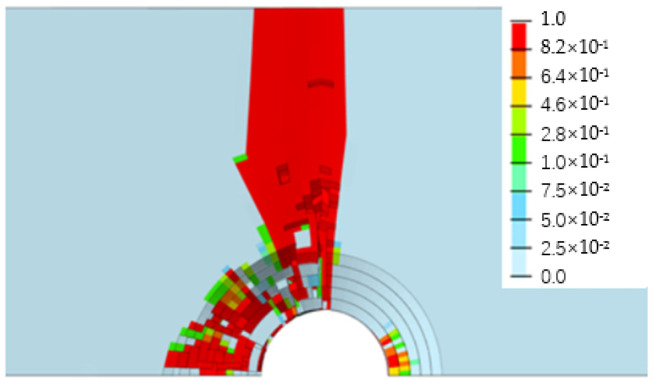
Matrix compression damage on the upper surface of the top plate.

**Figure 15 biomimetics-08-00215-f015:**
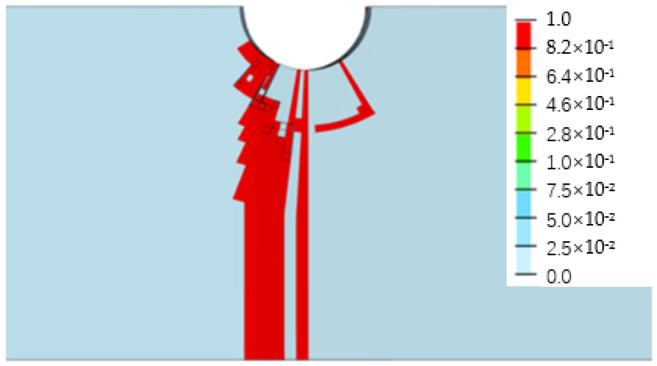
Fiber tensile damage on the bottom surface of the bottom plate.

**Figure 16 biomimetics-08-00215-f016:**
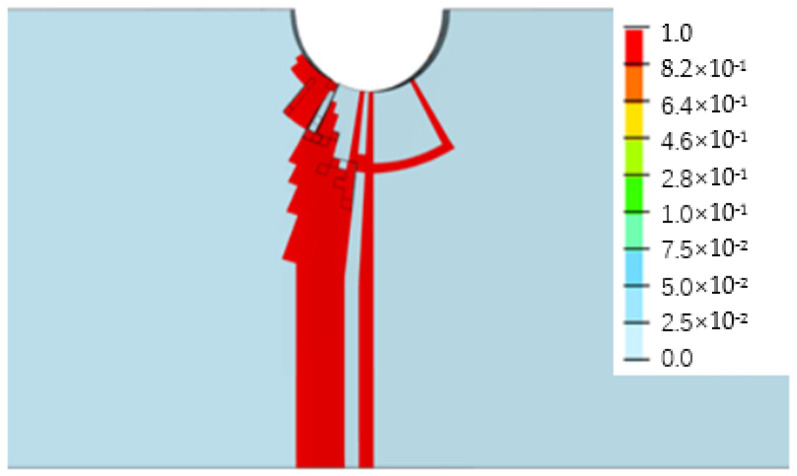
Matrix tensile damage on the bottom surface of the bottom plate.

**Figure 17 biomimetics-08-00215-f017:**
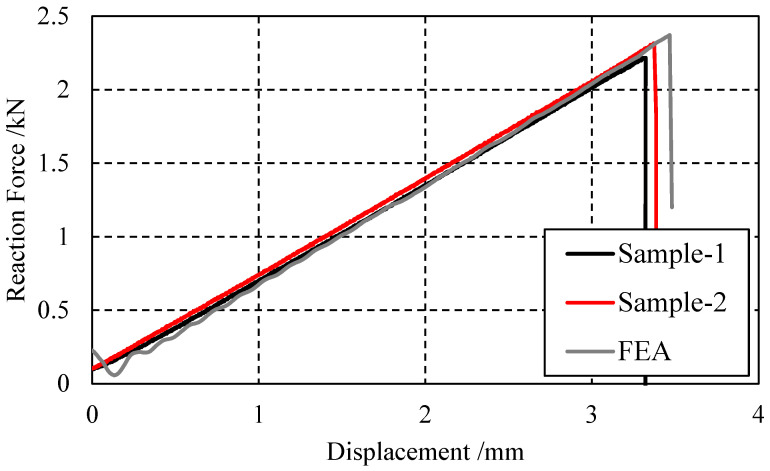
Reaction force–displacement curve found by testing and simulation.

**Figure 18 biomimetics-08-00215-f018:**
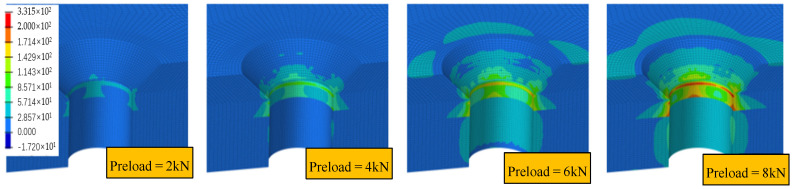
Maximum principal stress distribution under preload.

**Figure 19 biomimetics-08-00215-f019:**
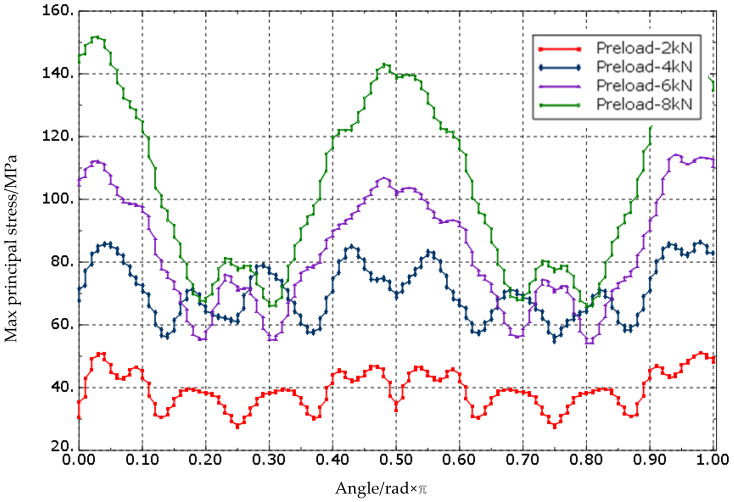
Maximum principal stress distribution alone hole perimeter direction.

**Figure 20 biomimetics-08-00215-f020:**
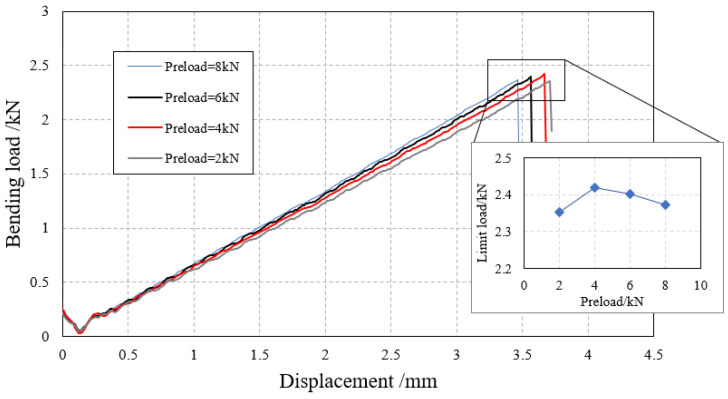
Displacement–bending load curve.

**Table 1 biomimetics-08-00215-t001:** Test specimen.

Parameter Name	Weaving Angle (°)	Number of Plies	Manufacturing Conditions	Size	The Total Thickness
Numerical value	0/90	30	High pressure in an autoclave at 0.5 MPa and 120 °C	130 mm × 30 mm × 6 mm	3.0 mm

**Table 2 biomimetics-08-00215-t002:** Material properties of countersunk head bolts (Unit: N-s-MPa-mJ).

	E	μ	σ_b_	α_t_
**Countersunk bolt**	110,000	0.29	950	1.2000 × 10^−5^
**Gasket & nut**	210,000	0.3	235	

**Table 3 biomimetics-08-00215-t003:** Material properties of T300BZ-3234 (Unit: N-s-MPa-mJ).

Yang’s modulus-11	69,000	Longitudinal tensile strength (MPa)	756
Yang’s modulus-22	69,000	Longitudinal compression strength	557
Yang’s modulus-33	8300	Lateral tensile strength	756
Poisson’s ratio-12 plane	0.064	Lateral compression strength	557
Poisson’s ratio-13 plane	0.064	Longitudinal shear strength	118
Poisson’s ratio-23 plane	0.32	Lateral compression strength	118
Shear modulus-12 plane	4200	Longitudinal tensile fracture energy	45
Shear modulus-13 plane	4200	Longitudinal compression fracture energy	0.6
Shear modulus-23 plane	4200	Lateral tensile fracture energy	45
		Lateral compression fracture energy	0.6

**Table 4 biomimetics-08-00215-t004:** Friction coefficient of each contact pair.

Composite–composite	0.3
Bolt–composite	0.1
Bolt–gasket	0.15
Gasket–composite	0.1

## Data Availability

Where no new data were created, or where data are unavailable due to privacy or ethical restrictions. Informed consent was obtained from all subjects involved in the study.

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
