# Peer review of "Ultimate Strength Study of Structural Bionic CFRP-Sinker Bolt Assemblies Subjected to Preload under Three-Point Bending"

_biomimetics, 2023, doi:10.3390/biomimetics8020215_

Round 1

Reviewer 1 Report

The paper describes a numerical analysis of CFRP-countersunk bolt assembly based on the Hashin failure criterion. The model was further benchmarked with experimental results. The paper is well-written and the results are interesting. However, the biomimetic aspect is assent, and the correlation between water bears and bolt assembly preload is unclear/absent. Biomimetics is typically a design approach in which specific natural features are investigated and then reproduced artificially to imitate biological performances. In my opinion, this aspect is absent and, due to that, it could be more appropriate for other journals.

Overall, the following aspects should be improved:

1.    On line 150, it is not clear the number of the figure you are talking about;

2.    On line 185, please remove “the test”, the sentence “to test pieces” is clear enough;

3.    On line 209, please specify the kind of material, the commercial code should be reported in parenthesis with the producer name;

4.    The sentence in lines 211-214 (“Drill a hole…bolt”) should be reformulated. The authors seem to give instructions instead of describing the procedure followed for sample fabrication;

5.    In Section 3, please specify the software employed for numerical simulation.

6.    Figure 16 reference in the text is missing

Author Response

Point 1: On line 150, it is not clear the number of the figure you are talking about;

Response 1: Thanks, it’s a manuscript error and we have fixed it in the recent concept.

Point 2: On line 185, please remove “the test”, the sentence “to test pieces” is clear enough;

Response 2: Thanks for advice, we have already fixed the error.

Point 3: On line 209, please specify the kind of material, the commercial code should be reported in parenthesis with the producer name;

Response 3: We have already labeled the producer name based on your suggestion.

Point 4: The sentence in lines 211-214 (“Drill a hole…bolt”) should be reformulated. The authors seem to give instructions instead of describing the procedure followed for sample fabrication;

Response 4: We have already reformulated the sentence.

Point 5: In Section 3, please specify the software employed for numerical simulation.

Response 5: We have already specify the software in the manuscript.

Point 6: Figure 16 reference in the text is missing

Response 5: We have fixed some error in the issue. In line 332 of original manuscript, the correct figure is “Figure 15” instead of “Figure 10”. In line 336, it’s “Figure 16” instead of “Figure 11”.

Reviewer 2 Report

The paper is focused on the finite element analysis of countersunk bolted joints including the ultimate load determination. Experimental results accompany the investigation. Overall, different aspects of the theoretical and numerical approaches need to be improved. The revised version of the manuscript should solve the following issues:

·         Paragraph 2.1.1, this section should be completed with a further technical description of the characteristics presented by the in-house developed experimental apparatus to give a full understanding to the reader. The addition of a Figure is useful to the scope.

·         Figures 1 - 6, these figures should show real photographs of the machine components together with the relative CAD view.

·         Paragraph 2.1.3, consider the following aspects:

o   Specify that two specimens were realized for the test.

o   Add a Figure showing the complete geometry of the specimen.

o   Insert a table with the material properties of the specimen, the material properties should be reported explicitly, the mention to the reference in Paragraph 3.3 is not sufficient.

·         Line 240, no equation is shown above this line.

·         Paragraph 3, much information about the finite element model is lacking. The following details are necessary and should be included: typology and order of the elements, mesh convergence analysis, presence of contact elements, and boundary conditions. Also, a Figure of the mesh should be included.

·         Line 272, the finite element model is not an analytical model but a numerical one.

·         Lines 288-294, this part is not clear since the model and the mentioned coordinate system are not shown.

·         Line 305, the results of the analysis are not analytical.

·         Figures 10-13, these Figures cannot be used as experimental verification of the model because they cannot be compared to Figure 8(a) which shows a lateral view of the specimen. The comparison of the model and experimental results should be more precise, and a close view of the specimen should be shown with the one of the FE model.

·         Figure 14, explain why the FEA curve is wavy in the initial stage.,

·         The scientific literature about the analysis of bolted joints is wide and differentiated; thus, the references should be enlarged with more works, such as: https://doi.org/10.1016/j.compstruct.2020.112005; https://doi.org/10.1016/j.prostr.2020.02.078; https://doi.org/10.1016/j.compstruct.2020.112770

Author Response

Point 1: Paragraph 2.1.1, this section should be completed with a further technical description of the characteristics presented by the in-house developed experimental apparatus to give a full understanding to the reader. The addition of a Figure is useful to the scope.

Response 1: Thanks for your suggestion, and we have followed your advice in the recent manuscript.

Point 2: Figures 1 - 6, these figures should show real photographs of the machine components together with the relative CAD view.

Response 2: In the recently manuscript, we changed figure 8 into 2 photograph, (a) is the fixture, and (b) is the installation.

Point 3: Paragraph 2.1.3, consider the following aspects:

3.1 Specify that two specimens were realized for the test.

Response : Thanks, we have already specified.

3.2 Add a Figure showing the complete geometry of the specimen.

Response : We have already add a figure showing the specimen.

3.3 Insert a table with the material properties of the specimen, the material properties should be reported explicitly, the mention to the reference in Paragraph 3.3 is not sufficient.

Response : We have already inserted material properties tables.

Point 4: Line 240, no equation is shown above this line.

Response 4: We have already fixed this errors.

Point 5: Paragraph 3, much information about the finite element model is lacking. The following details are necessary and should be included: typology and order of the elements, mesh convergence analysis, presence of contact elements, and boundary conditions. Also, a Figure of the mesh should be included.

Response 5: OK, we have reformulated the mesh information, and add a mesh figure as figure 11.

Point 6:  Line 272, the finite element model is not an analytical model but a numerical one.

Response 6: We have already fixed this errors.

Point 7: Lines 288-294, this part is not clear since the model and the mentioned coordinate system are not shown.

Response 7: We have already reformulated these sentences.

Point 8: Line 305, the results of the analysis are not analytical.

Response 8: We have already fixed this errors.

Point 9: Figures 10-13, these Figures cannot be used as experimental verification of the model because they cannot be compared to Figure 8(a) which shows a lateral view of the specimen. The comparison of the model and experimental results should be more precise, and a close view of the specimen should be shown with the one of the FE  model.

Response 9:  We changes figure 10(a) into two top view photograph.

Point 10: Figure 14, explain why the FEA curve is wavy in the initial stage.,

Response 10: When using explicit finite element method for analysis, curve oscillations may occur when the loading rate is fast (such as in three-point bending test specimens). This is because the time steps used in the algorithm of explicit finite element method are relatively short and the computer needs to complete a large number of calculations in a short period of time. If the time step used is too short, numerical dissipation and diffusion errors will occur, leading to oscillations in the curve. In addition, this phenomenon may also be related to the nonlinearity and instability of the material itself.

Point 11: The scientific literature about the analysis of bolted joints is wide and differentiated; thus, the references should be enlarged with more works, such as: https://doi.org/10.1016/j.compstruct.2020.112005; https://doi.org/10.1016/j.prostr.2020.02.078; https://doi.org/10.1016/j.compstruct.2020.112770

Response 11: We have already added some more references in the recent manuscript.

Reviewer 3 Report

In this paper, it is investigated the load distribution in the bolt holes and the effect of different preloads on the bending strength of the joints. And the 3-point bending fatigue test was selected as the testing method for this study.

1)     To show and explain fiber and matrix failure patterns of the countersunk bolt-CFRP assemblies in the 3-point bending fatigue test, tensile/compression?

2)     To supplement the finite model of the countersunk bolt-CFRP assemblies, including the boundary conditions;

3)     To list the material properties of the head bolts, gasket and composite laminate used in the study, And does they changed in the range of large temperature?

4)     Due to the countersunk bolt-CFRP assemblies just in the middle of the model and its elastic failure patterns, is only half of the entire model considered in the longitudinal direction and symmetric boundary conditions applied to the symmetric plane rational? To explain that considering the results of figures 10 to 13.

5)     To mark the curves of different preload in fig. 16.

Author Response

Point 1: To show and explain fiber and matrix failure patterns of the countersunk bolt-CFRP assemblies in the 3-point bending fatigue test, tensile/compression?

Response 1: In another of our papers we reveal the form of fatigue damage of fibers and substrates

Point 2: To supplement the finite model of the countersunk bolt-CFRP assemblies, including the boundary conditions;

Response 2: OK, we have reformulated the mesh information, and add a mesh figure as figure 12.

Point 3: To list the material properties of the head bolts, gasket and composite laminate used in the study, And does they changed in the range of large temperature?

Response 3: We have already listed the material properties related to the FE model in Tab2 to Tab4.

And the other question, my understanding is that your question is mainly about whether temperature changes when preload is applied affect the material properties of the model, in Section 4.1, line 328. My answer is that there is no effect on our result. In this numerical analysis of the FE model in step-1, the preloading of the bolt is supplied by the temperature cooling (as line 328), which means that we only set a coefficient of thermal expansion in the bolt material. Yong’s modulus is a constant and it is not related to the temperature. And temperature only changes in the bolt node set in the model.

Point 4: Due to the countersunk bolt-CFRP assemblies just in the middle of the model and its elastic failure patterns, is only half of the entire model considered in the longitudinal direction and symmetric boundary conditions applied to the symmetric plane rational? To explain that considering the results of figures 10 to 13.

Response 4: In fact, we have simplified the numerical model to a certain extent to save the calculation time. The advantages of using a symmetric model in finite element calculations are that it can simplify complex geometries into simpler ones, thereby reducing the complexity of finite element calculations. Specifically, for structures or components with axial symmetry, the entire geometry can be described by only one sectional model and a certain number of circumferential elements, which eliminates the tedious process of geometry modeling and meshing. In addition, using a symmetric model can also improve the accuracy and precision of calculations, as it utilizes the symmetry of the geometry, reduces errors between adjacent physical regions, and increases the reliability of the calculation results. The advantage of using the full model in finite element analysis compared to using a symmetric model is that it can more accurately predict the stress and deformation of the structure, because the full model considers all geometric details and material non-uniformities of the structure. However, this increases the computation time and cost. In contrast, the symmetric model ignores certain details of the structure but has lower computation time and cost. The choice of which model to use depends on the specific situation and requires a balance between accuracy, efficiency, and cost.

Point 5: To mark the curves of different preload in fig. 16.

Response 5: We have fixed this error by a legend.

Round 2

Reviewer 1 Report

In my opinion, the paper is off-topic. 

Reviewer 2 Report

The required amendments were considered and implemented.

The paper can be accepted in present form.

Reviewer 3 Report

The manuscript has been sufficiently improved for publishing.